# Automated Monitoring of Human–Computer Interaction for Assessing Teachers’ Digital Competence Based on LMS Data Extraction

**DOI:** 10.3390/s24113326

**Published:** 2024-05-23

**Authors:** Eduard de-Torres, Xavi Canaleta, David Fonseca, Maria Alsina

**Affiliations:** Human-Environment Research Group, Universitat Ramon Llull (URL), La Salle, 08022 Barcelona, Spain; eduard.detorres@salle.url.edu (E.d.-T.); xavier.canaleta@salle.url.edu (X.C.); maria.alsina@salle.url.edu (M.A.)

**Keywords:** automation, human–computer interaction, digital literacy, digital competence, 21st century skills, teacher evaluation

## Abstract

The fast-paced evolution of technology has compelled the digitalization of education, requiring educators to interact with computers and develop digital competencies relevant to the teaching–learning process. This need has prompted various organizations to define frameworks for assessing digital competency emphasizing teachers’ interaction with computer technologies in education. Different authors have presented assessment methods for teachers’ digital competence based on the video analysis of recorded classes using sensors such as cameras, microphones, or electroencephalograms. The main limitation of these solutions is the large number of resources they require, making it difficult to assess large numbers of teachers in resource-constrained environments. This article proposes the automation of teachers’ digital competence evaluation process based on monitoring metrics obtained from teachers’ interaction with a Learning Management System (LMS). Based on the Digital Competence Framework for Educators (DigCompEdu), indicators were defined and extracted that allow automatic measurement of a teacher’s competency level. A tool was designed and implemented to conduct a successful proof of concept capable of automating the evaluation process of all university faculty, including 987 lecturers from different fields of knowledge. Results obtained allow for drawing conclusions on technological adoption according to the teacher’s profile and planning educational actions to improve these competencies.

## 1. Introduction

The evolution of digital technology has had a global impact across all social and professional spheres. Nowadays, mechanisms of communication, socialization, work, or leisure have been overrun by the adoption of new technologies. These new technologies modify behavior patterns, enhancing the adoption of the knowledge society. The constant improvements in Information and Communication Technologies (ICT) are enabling the digitalization of many professional sectors, either supporting or replacing some traditional tasks. However, this digitalization process has increased the interaction between humans and computers, as a direct consequence of the increase in available digital tools, resources, and processes. Therefore, this innovation process has been accompanied by the necessity to develop new skills and competencies that allow for effective human–computer interaction. In the education sector, these changes have had a significant impact on how students, educational institutions, and faculty interact and perform their functions [1]. The teacher in the knowledge society must be able to interact with computers in a way that supports and enhances the teaching–learning processes.

These new skills necessary to create meaningful interactions with computers are known as digital literacy [2,3,4,5,6] or digital competence [7,8,9]. These terms have been established in the scientific community to explain the need for society to introduce new fundamental skills for the proper development of daily tasks. While the concept of digital literacy has a broader usage in the United Kingdom, the United States, and Asia, it encompasses a more general use of digital skills; on the other hand, the term digital competence has been more widely adopted in regions of continental Europe and South America and is prevalent in educational environments [10].

Human interaction with these digital technologies in educational contexts has allowed for teachers’ improved productivity, in terms of class administration, content and material management, or digital assessment [11,12,13,14,15]. Downloading a list of students, emailing homework, or answering questions online are only a few examples. However, educational innovation takes place when teachers create new processes to enhance teaching and learning [15]. Years ago, access to information required access to physical spaces, such as libraries, but nowadays, teachers and students can interact with computers in many ways to search for information that can be found via several platforms, which boosts shared learning and project-based or research-based methodologies [11,12,13,14,15,16]. Yet teachers may be reluctant to embrace change if they do not have the skills to interact correctly with digital technology, either because of poor design in the usability and user experience of the technology, or because they lack digital competence [11,12,15,16,17,18].

In recent years, the COVID-19 pandemic posed an important test for the education sector, which had to accelerate the digitalization process [19,20,21] to respond to the emergency measures placed by institutions. In addition, the rise of artificial intelligence with the introduction of ChatGPT in late 2022 is challenging the role of the teacher in the face of the potential for new interactions between students and AI [22,23,24,25].

However, this digitalization process had already begun years before with the emergence of models and frameworks for assessing digital competence. The initial models focused on assessing the digital competence of citizens or students, while later frameworks emerged for assessing teachers’ digital competence (TDC). In the section below, the frameworks most relevant in the literature to assess TDC will be presented.

### 1.1. Frameworks for the Assessment of Teachers’ Digital Competence

Among the most well-known frameworks, Mishra and Koehler [26,27] presented their Technological Pedagogical Content Knowledge (TPACK) framework in 2006. TPACK is a conceptual framework that expands upon Shulman’s construct [28] known as Pedagogical Content Knowledge (PCK) by integrating technology into education. TPACK defines three different types of knowledge that all educators should develop in order to create meaningful learning experiences. The first knowledge, known as Content Knowledge, defines the up-to-date knowledge of the teaching subject. Next, the Pedagogical Knowledge covers the pedagogical strategies concerning the effectiveness of the teaching process. Last, the Technology Knowledge, involves the digital learning recourses available to teachers and students, including teachings aids or digital ecosystems like Learning Management Systems (LMS). By combining the three Knowledges, teachers can develop first their Technological Content Knowledge, which involves the selection of the best digital tools and resources to support the learning content; and the Technological Pedagogical Content Knowledge, which is the ability to select tools depending on the learning objective and the broader conditions of the teaching processes. The goal of TPACK is for teachers to use their knowledge on human–computer interaction, along with pedagogical knowledge related to teaching, and knowledge on the content to be taught. The proper integration of these three areas of knowledge should allow teachers to define learning situations suitable for their students while using the best interaction with technology to enhance teaching–learning processes.

A second framework that comes across the literature in many countries is the UNESCO ICT Competency Framework for Teachers (ICT CFT) [29]. This framework was presented in 2008 and updated to its third version in 2018 [12]. It is based on the principle that the knowledge society is currently being developed on human interaction with information and communication technologies, aligned with the United Nations Sustainable Development Agenda. This framework emphasizes that teachers who have digital competencies to use ICT in their professional practice will be able to conduct high-quality learning experiences and foster the development of students’ ICT skills. ICT CFT consists of 18 competencies organized in six areas that represent the different aspects of teacher professional practice: (1) Understanding ICT in Education Policy; (2) Curriculum and Assessment; (3) Pedagogy; (4) Application of Digital Skills; (5) Organization and Administration; and (6) Teacher Professional Learning. These competencies define three successive levels (known as stages) to assess teachers’ acquisition of knowledge related to the pedagogical use of digital tools and resources. The framework presents elaborated and comprehensive objectives for each competency and provides examples of skills that would represent such competencies.

In the United States, in 2008, the International Society for Technology in Education (ISTE) also introduced its National Educational Technology Standards for Teachers (NETS-T) [14,30,31]. This framework promotes the creation of educational programs to improve teachers’ attitudes toward new technologies, as well as the creation of personalized rankings demonstrating digital competence. This framework was updated in 2021 and became the ISTE Standards: For Educators [13]. These standards are broken into four sections aimed at students, educators, education leaders, and coaches. As for the section pertaining to educators, it defines 24 characteristics organized in seven proficiencies. These characteristics allow educators to self-diagnose, and provide a roadmap to deepen their educational practice, promote collaboration with peers, rethink traditional approaches, and prepare students to drive their own learning.

In Europe, in 2006, the European Parliament presented a proposal for essential competencies for the lifelong learning of its citizens [32]. One of these competencies was related to digital technology and the way citizens interact with it. After presenting DigComp in 2013, a framework aimed at assessing digital competence in citizens [18], in 2017, the JRC Research Center of the European Union presented a framework for assessing digital competence in teachers [11]. This framework, known as the European Framework for the Digital Competence of Educators (DigCompEdu), recognizes that the interaction between teachers and computers is changing and, consequently, it is important for them to develop a set of more sophisticated digital competencies. The framework defines 22 competencies organized into six areas. Each competency has distinct indicators linked to learning and teaching activities and focuses on the development of the information society and the mix of pedagogy with technology.

In Spain, in 2017, INTEF presented the *Marco de Referencia de la Competencia Digital Docente* (MRCDD) [33], an adaptation to the Spanish context of the European DigCompEdu framework, and was updated in 2022. MRCDD consists of six areas and 23 competencies structured into six levels of development. Each of these competencies provides a detailed description as well as a set of level indicators and performance statements associated with each of the development levels.

These frameworks highlight the need for proficiency in digital skills to go hand in hand with pedagogical knowledge, in order to improve teaching–learning experiences. The main difference between TPACK and the other models is the fact that it focuses exclusively on the development of digital competencies in teachers, while the UNESCO framework, DigCompEdu, and NETS-T, emphasize that the teachers must also foster the development of digital skills in their students. The latter frameworks also have in common the way they are structured. They structure the competencies important for the educational practice into items that are organized in areas of expertise, define progression models, and suggest abilities to learn for improvement. Furthermore, they describe a series of practical activities at different levels to exemplify the skills necessary to achieve each level and aim to reinforce teachers’ reflexivity in the field of media pedagogy [31].

As observed, different organizations consider that using digital technology in the education sector is vital in the knowledge society. This requires teachers to develop new digital skills. Teachers must learn how to properly interact with digital technology to adapt teaching–learning processes.

Having reviewed the state of the art on the concepts of digital skills and digital competence, and after examining the most relevant frameworks in the literature that propose ways to assess the teachers’ digital competence, the following research question has been posed:RQ: Can teachers’ digital competence be assessed from teacher’s interaction with digital technologies?

### 1.2. Digital Competence Assessment through Human–Computer Interaction

In order to answer the research question previously stated, it was considered necessary to investigate models and instruments that assess digital competence, whether the proposals do so through interaction with digital technology or not. During the review of the literature and state of the art, it was revealed that different authors have different ways of approaching the assessment of the teachers’ digital competence. While authors working on assessing teachers’ digital skills typically use the term teachers’ digital competence, in other engineering forums focused on the development of these digital technologies, it is common to address the interaction between people and digital technology under the term human–computer interaction. To be able to compare some works from different areas, it is believed interesting to use the concept of teacher–computer interaction. This concept, similar to digital competence focused on teachers, would be responsible for addressing the meaningful interactions that teachers make when interacting with digital technology. After presenting this term, the three main approaches that have been found to assess the digital competency that teachers have when interacting with technology will be examined.

The first group of authors seeks to evaluate teachers’ interaction with technology, and therefore, their digital competency, by measuring its impact on students. Two main categories could be identified in this group: those aiming to measure students’ engagement levels using deep learning techniques, and those attempting to identify behavior patterns through electroencephalograms (EEG). Hu et al. [34] propose to optimize teaching processes by recognizing engagement patterns in students. Their proposal employs deep learning techniques to analyze classroom video recordings. These recordings are stored in a database and used to train a k-nearest neighbors (KNN) classifier algorithm capable of identifying different levels of engagement in a class. Similarly, Rathod et al. [35] propose a system based on a convolutional neural network (CNN) algorithm to analyze a bank of video recordings featuring children’s faces. The goal is to train a system to measure the level of engagement in a class. Wang et al. [36] present a similar proposal based on the use of an Extended-Efficient Layer Aggregation Network (E-ELAN) applied in Smart Classroom environments. On the other hand, Fuentes-Martinez et al. [37] propose an alternative solution to using cameras and microphones to record the classroom, by using low-cost EEG platforms. Their objective is to analyze beta bands obtained from EEG to determine a student’s attention level and provide feedback to the teacher. Similarly, Liu et al. [38] propose the use of EEGs combined with a support vector machine (SVM) classifier to measure students’ attention levels by comparing data obtained in two different scenarios where the student is paying attention or not.

Other proposals presented by a second group of authors use techniques similar to the previous ones but put the focus on evaluating the teaching provided by the teacher. Guo et al. [39] propose a system based on computer vision and intelligent speech recognition (ISR) that aims to analyze, by means of artificial intelligence techniques, the body language, teaching style, and speech of a teacher inside the classroom to generate an evaluation model.

Finally, a third group of authors base the evaluation of teachers’ digital competence on the analysis of indicators according to pre-established criteria in an assessment framework. According to the analysis conducted by Sillat et al. [40], a large number of these works propose a quantitative methodology for the definition of tests or questionnaires, where the most commonly used framework is DigCompEdu [17,41,42,43,44,45,46]. These authors share the goal of creating a tool capable of assessing teachers’ digital competence, as defined in the European framework DigCompEdu [11]. However, it is worth noting that other authors propose similar solutions based on other international frameworks. Mengual-Andrés et al. [47] suggest combining different frameworks such as the ISTE Standards for Educators [13] and the UNESCO ICT Competency Framework for Teachers [12] to develop an international digital competence assessment questionnaire. A third approach to the use of assessment frameworks considers it important to be able to measure the level of competence of teachers in training. In the review conducted by Røkenes and Krumsvik [48], they analyze different training programs designed to teach future secondary education teachers. This review assesses the quality of training programs considering the level of digital competence of teachers in training based on the knowledge demonstrated by students regarding the use and interaction with digital technologies applied to teaching.

### 1.3. Proposed Model

After reviewing the different applications found in the literature, the objective of this article is to present a model for assessing teachers’ digital competence that considers the characteristics of the different proposals presented earlier. The model (1) will be based on the European framework for assessing digital competence, DigCompEdu, as it is the approach proposed by a large part of the literature; (2) will be focused on the interaction that teachers have with computers and digital technology, without assessing their impact on students; and (3) will seek to automate the process of analyzing predefined metrics, without using video or audio recordings to comply with the European General Data Protection Regulation (GDPR).

### 1.4. European Framework for the Digital Competence of Educators: DigCompEdu [11]

Once the state of the art has been presented and the proposal established, we will proceed to further detail the characteristics of the chosen assessment framework. As can be seen in Figure 1, this framework defines a total of 22 competencies that every teacher should develop, grouped into six competency areas.

*Professional Engagement* is directly focused on the professional development of the teacher and their interaction with digital technology in work environments. Technology is evaluated based on its professional usage regarding colleagues, students, and families.*Digital Resources* focuses on the necessary skills to create, share, and manage digital resources for teaching. It emphasizes responsible and effective interaction with technology and the licenses that derive from it.*Teaching and Learning* emphasizes skillfully orchestrating the incorporation of digital technologies throughout the diverse phases of teaching and learning processes. It also advocates for embracing student-centered approaches and methodologies, fostering a transformative shift in instructional practices.*Assessment* is concerned with the interaction with digital tools and digital strategies during the evaluation and refinement of teaching–learning processes. Leveraging these technologies enhances existing assessment methods and introduces novel approaches to evaluation.*Empowering Learners* highlights the capability that digital technologies can provide to tailor learning activities that adapt to each student’s level, interests, and needs. In addition, digital technologies applied to education have the capability of fostering active participation of students in the process of learning.*Facilitating Learner’s Digital Competence* is concerned with the necessary skills and competences that teachers must have to facilitate student acquisition of digital competence.

Each competency features a detailed progression system based on six competency levels similar to those defined in the Common European Framework of Reference for Languages (CEFR), as depicted in Figure 2.

*Newcomer (A1)*: Teachers who recognize the potential of interacting with digital technologies to enhance pedagogical and professional practice but have had very little contact with these technologies.*Explorers (A2)*: Teachers who recognize the potential of digital technologies and are interested in improving pedagogical and professional practice with their usage. Although they have begun to explore some areas of digital competence, they need guidance and support to move forward more consistently and comprehensively.*Integrators (B1)*: Teachers who interact with various digital technologies and incorporate them into their professional practices creatively. They are independent; however, they need more time to understand which tools work best in which contexts and to adjust them to pedagogical strategies.*Experts (B2)*: Teachers who interact with various digital technologies securely, creatively, and critically to enhance their professional activities. They deliberately select digital technologies for particular situations and seek to understand the advantages and disadvantages of different digital strategies.*Leaders (C1)*: Teachers who have a comprehensive and effective approach to interacting with digital technologies to enhance pedagogical and professional practices. They rely on a wide range of digital strategies and act as sources of inspiration for others.*Pioneers (C2)*: Teachers who question the adequacy of current digital and pedagogical practices and are leaders in this field. They drive innovative interactions and experiment with recent digital technologies to develop new pedagogical approaches.

After introducing the state of the art of teachers’ digital competence, presenting the proposed model to be developed, and detailing the evaluation framework used, an explanation of the research methodology will follow, together with the objectives and limitations, followed by the presentation and analysis of the results, their discussion, and the conclusions.

## 2. Materials and Methods

For the development of this proposal, an Instructional Design [49] has been followed using the ADDIE model [50,51]. This methodology has resulted in a quantitative study of the digital competence of university faculty.

The ADDIE methodology is an iterative model used in development research that defines 5 phases: Analysis, Design, Development, Implementation, and Evaluation. During the analysis phase, the team should consider the learning environment, the audience, the problem to solve, and the limitations. It is important to identify these key concepts to set the goals, scope, and constraints. Once the analysis is complete, the team should proceed to the design phase. Through the design phase, the team should map the process to achieve the desired outcome and decide the feedback mechanisms. In this phase, the team should collect data to inform the decision-making process, if necessary. During the development phase of the ADDIE model, the team should create any asset or material defined in the previous phase. The implementation phase deals with the delivery of the research. The final phase is Evaluation when the team measures the effectiveness and efficiency of the research.

### 2.1. Objectives, Scope, and Constraints

During the analysis phase of the ADDIE model, an analysis of the problem has been undertaken. This analysis allowed the definition of the objective, the scope, and the limitations that must be considered. 

The objective of this work is to analyze the level of digital competence in university faculty members through automatization techniques. The analysis must be based on faculty interaction with the Learning Management System (LMS) provided by the university. The results obtained must allow experts to determine the need to implement training actions that provide faculty with the necessary tools to effectively perform their teaching profession.

Since the experimentation will be conducted in a university in Spain, a member state of the European Union, the chosen evaluation framework will be one specific to this region. Therefore, to assess the level of digital competence of faculty members, the European framework for the digital competence of educators, DigCompEdu, will be used to determine the competency levels through its different areas. Although Spain has presented the MRCDD framework for the evaluation of TDC, it has been decided to use the parent framework, DigCompEdu, to allow further research to be expanded outside of the Spanish context. 

The study population has been limited to faculty members of the university, who have user account in the university’s LMS and are working in any of the four schools (Architecture, Business, Engineering, and Philosophy) teaching in Bachelor’s or Master’s programs.

For this first iteration, the analysis of teachers’ digital competence will be restricted to their interaction with the LMS, although there are other digital tools capable of complementing the information that can be obtained.

### 2.2. Metrics and Indicators

During the design phase of the ADDIE model, the indicators necessary to measure teachers’ digital competence based on their interaction with an LMS have been defined.

To define the metrics and indicators, the Expert Judgment methodology has been used. The university has the Department of Deployment of Teaching Methodologies and Policies (DDMPD, according to its acronym in Catalan), responsible for managing the academic and pedagogical aspects of the university’s LMS. This department consists of a team of experts in digital technology applied to teaching, as well as experts in the Moodle platform, which is the LMS used by the university.

Based on their knowledge and expertise, all activities and resources available in the LMS have been analyzed. This analysis included those resources common to any default Moodle installation but also those plugins locally installed in the university’s LMS.

The result of this analysis, and the indicators and metrics defined during the process, can be seen in Table 1 and Table 2. In total, 55 activities and resources were analyzed, 29 of them available by default in any Moodle installation (Table 1). Additionally, 26 plugins installed in the university’s LMS were analyzed (Table 2) to complement the study.

To create the indicators for each activity and resource, the experts analyzed the definition of each competency described by DigCompEdu, as well as the proposed examples and uses provided by the framework. Based on this knowledge, and taking into account the uses and characteristics of each activity and resource, a competency level was determined for teachers using those activities and resources in their courses. The indicators use the competency levels presented in the DigCompEdu framework (A1, A2, B1, B2, C1, and C2) to describe the level of engagement in the user’s interaction with the digital tool, to assess the teachers’ digital competence. Thus, a set of indicators capable of assessing a teacher’s level of digital competence based on their interaction with an LMS such as Moodle can be created.

The result of the design phase was 237 indicators capable of assessing 17 of the 22 competencies defined by DigCompEdu. Some competencies were deemed as non-assessable due to the nature of an LMS, which cannot represent all the professional competencies defined by DigCompEdu.

Special mention should be made of some activities or resources marked with an X in Table 1. These are tools capable of embedding a variety of external elements. The experts considered that a competency level could not be standardized for those cases and marked them with this special category to be studied in further detail in the future.

### 2.3. Development of the Extraction Tool

The third phase of the ADDIE model involves the development of all the necessary materials and resources to accomplish the implementation phase. In this phase, a knowledge extraction software was developed capable of automating the analysis of all the information stored in the LMS database required to evaluate the indicators defined in the previous stage.

The instrument was created with two working modes: a first mode capable of analyzing the information of a single teacher to visually display their competency levels in each of the 22 competencies of DigCompEdu; and a second mode designed to analyze massively the data of all users in the LMS database installed on the university’s servers. This second mode presents the results in a tabular format pseudo-anonymizing the information, to allow the identification of the teacher by their user ID. While the first mode has a pedagogical objective focused on informing the teacher of their current knowledge situation in order to advise them of their strengths and weaknesses, the second mode seeks a cross-sectional analysis of the faculty to detect widespread deficiencies that may require action by the experts in the DDMPD department.

In Figure 3, the results view developed for the first mode of operation is displayed. It is important to remember that the target user of this tool may not be familiar with the evaluation model used and may not be competent in analyzing the obtained data. For this reason, it has been decided to use radar charts, widely known in competency analysis comparison, to visually and graphically display the different levels demonstrated by the teacher. Additionally, the information is displayed again in table form to allow for a second analysis of the information.

The experiment was conducted based on the data generated by the interaction of the university’s faculty with the university’s LMS (Moodle). All data had been stored in the form of logs that were analyzed once the 2022–2023 academic year concluded.

The population analyzed during this experiment consists of all university professors of Bachelor’s and Master’s programs who work full-time or part-time and teach in face-to-face, online, or hybrid programs. Initially, 1303 users with the role of “editing-teacher” were obtained. This role is used within the LMS to identify users with editing permissions corresponding to the role of a teacher.

During this initial iteration, 7 competencies from various competency areas were deliberately selected to conduct a proof of concept and validate the correctness of our hypothesis. Among the 22 competencies delineated within the DigCompEdu framework, all assessable competencies from areas 1 to 3 were included for examination.

### 2.4. Limitations and Constraints

During the development of the instrument to conduct the experimentation, several limitations were detected.

During the analysis of the initial study population, it was detected that some users, despite having the appropriate role for a teacher, did not perform teaching tasks and had to be removed from the study population. These removed users belonged to one of the following categories: (1) users from administration and support departments that support courses or programs and need editing permissions; (2) students hired by the university for academic support tasks; (3) instructors of in-company courses that are not part of regulated Bachelor’s or Master’s programs; and (4) instructors of specific online Master’s programs following the Self-Directed Based Learning methodology (SDBL) [52], which requires that LMS activities and resources be designed and managed by support staff. Therefore, teachers in these programs cannot be evaluated because the interaction with the LMS in their courses is not theirs, but done by third parties on their behalf. Once the results were filtered, 316 users belonging to any of the four categories mentioned above were removed. Thus, the final analyzed population consisted of 987 users.

While considering the competencies to incorporate in the initial iteration, it was decided to include all assessable competencies from areas 1 to 3. During the design phase, the experts from the DDMPD department who defined the indicators determined that some competencies like *C1.2-Professional Collaboration*, *C1.3-Reflective Practice*, and *C1.4-Digital CPD* from Area 1 could not be assessed. These competencies examine the capacities of educators to interact with digital tools that enhance their professional development as workers. They do not directly relate to teaching or learning and cannot be appraised within an LMS, an environment focused on teaching–learning processes. On the other hand, competency *C2.1-Selecting Digital Resources* assesses the faculty’s aptitude to discriminate and opt for digital tools and resources tailored for teaching and learning, in accordance with learning objectives, contextual distinctions, pedagogical methodologies employed, and student demographics. This analytical and introspective effort remains beyond the scope of this analysis, as the decisions made by the teachers when selecting the digital resources before their interaction with the LMS do not leave discernible traces within the system logs that could be analyzed.

The analysis conducted by the team of experts from the DDMPD department determined that not all competency levels could have indicators. The competency level *Newcomer (A1)* in competency C2.3 and competencies from C3.1 to C3.4, as well as the level *Explorer (A2)* in competency C2.3, are unverifiable levels because the interaction with the LMS and its activities and resources, according to the criteria established in the DigCompEdu framework, already allows for confirming higher competency levels than those established at levels A1 or A2. For example, competency *C2.3-Managing, Protecting, and Sharing Digital Resources*, defines interaction with the professor at level A1 as “*I store and organize digital resources for my own future use.*” [11], and level A2 as “*I share educational content* via *e-mail attachments or through links.*” [11]. Upon reaching level B1, we find that sharing digital resources in an LMS already equips educators with this level of competency, so any educator who interacts with an LMS to share educational materials with their students already demonstrates a competency level equivalent to B1.

Related to the initial study conducted by the expert group, limitations were also detected in assessing the highest competency levels. The *Leader* (C1) and *Pioneer* (C2) levels are often achieved by demonstrating the ability to reflect on current forms of interaction with digital tools and resources applied to teaching, and to develop new ones, promoting innovation and the creation of new proposals. Since this is a quantitative study, one of the limitations identified consists of the impossibility of detecting the teacher’s intent when interacting with technology. The quantitative analysis allows for obtaining objective results of the actual interaction of the teacher with digital tools and resources, but this analysis cannot detect the reflective thinking and decisions made that have led the teacher to choose this interaction. Therefore, there are no indicators in any competency that allow obtaining the *Pioneer* (C2) level. On the other hand, the *Leader* (C1) level includes in its definition teachers who demonstrate having a wide range of tools and resources with which they interact, to choose those most suitable for each moment and learning situation. According to this definition, competencies C2.2, C2.3, and C3.1, among those studied, allow defining indicators that assess this competency level, since the chosen LMS offers the possibility of interaction with a wide range of tools and resources. Again, quantitative analysis does not allow us to determine if the teacher’s uses are appropriate and well-reflected, as it can only rely on the quantity and type of elements used.

## 3. Results

Now that the methodology used has been explained, and the indicators and analysis tools have been presented, the obtained results will be discussed.

Table 3 shows the results obtained after analyzing the 987 users with the instrument developed. 

Each row in Table 3 displays the different competency levels defined by the DigCompEdu framework. An additional level, *Uninitiated (A0)*, was added to represent all those educators who have not been able to demonstrate minimum competency level based on their interaction with the LMS and the records left in the analyzed system logs.

Figure 4 depicts a stacked bar chart illustrating the information in Table 3 to facilitate a quicker visual comparison. On the other hand, Table 3 allows us to see more clearly the competence levels that do not have results since there are no indicators capable of assessing these levels. Those non-assessable levels due to the lack of indicators are marked with a hyphen (-) in the results.

In order to analyze the data comprehensively, a statistical study of the results obtained was conducted. Firstly, to process the results in numerical format, it was necessary to convert the competency levels from textual values to numerical values. To do so, values from 0 to 6 were assigned to the different competency levels between A0 and C2. Once this transformation was done, the mean (µ), standard deviation (σ), and mode (Mo) of the results were calculated to obtain aggregated information that allows us to draw conclusions from the results.

Table 4 displays the result of the statistical analysis, while Table 5 shows the results re-transformed according to the textual levels defined in the DigCompEdu framework to facilitate the interpretation of the results.

The results show that competencies in *Area 2*—*Digital Resources*, are the ones where teachers achieve better results. These two competencies are responsible for assessing the use of digital resources that support learning objectives and the teacher’s style. While competency C2.2 is responsible for managing the creation and modification of all types of digital resources, whether intrinsic to the LMS or external to it, competency C2.3 is responsible for evaluating the management of these resources in order to make them available to students. The mode (Mo) for these competencies has reached the levels of *Integrator* (B1) and *Expert* (B2), which determine that users do not depend on third parties during the process of creation, modification, and management of the resources used. To progress to competencies equivalent to the *Leader* (C1) or *Pioneer* (C2) levels, teachers should develop skills to be able to perform a deeper reflection on the available resources. Regarding competency C2.2, the results clearly show that the majority of the teachers (53%) demonstrate a level equivalent to B1, which describes teachers as capable of creating and modifying digital resources that integrate animations or interactive elements and adapt the use of resources to educational practices through interaction with multiple different resources. On the other hand, competency C2.3 is distributed among levels A0 (20%), B1 (33%), and B2 (44%), with only 3% of teachers demonstrating competency levels equivalent to C1. Competency C2.3 is strongly linked to interaction with tools that allow the management of digital resources, and DigCompEdu considers that interaction with tools like LMS is sufficient to demonstrate a non-dependent interaction level equivalent to *Integrator* (B1). Finally, in competency C2.3, it is remarkable that 194 teachers (20%) are unable to demonstrate interaction capabilities with the LMS corresponding to the management and sharing of digital resources. After analyzing these 194 LMS users more deeply to determine possible reasons for this low level, it was found that 94 (9.5% of the total population) of these users fail to demonstrate any competency through the interaction they have had with the LMS, suggesting that their interaction has been null during the academic year 2022–23. One hypothesis is that these 94 users may be teachers who have had a relationship in the past with some subjects of the programs studied but have not taught during the academic year 2022–23. Further analysis of the set of users is necessary to confirm this hypothesis.

The majority (69%) of the results obtained in competency C3.1 are between levels A2 (39%) and B1 (30%). These levels correspond to interaction with different types of tools and resources ranging from digital whiteboards and remote meeting rooms to the incorporation of digital devices by students. No user has achieved the maximum assessable level (*Expert B2*), which evaluates interaction with technologies that enhance chosen pedagogical strategies.

The results for competency C1.1 show that nearly half (46%) of the analyzed teachers fail to achieve a minimum level of competence and receive a qualification of *Uninitiated* (A0). Competency C1.1 is responsible for demonstrating that teachers are able to interact with digital tools to communicate relevant information to their students regarding the organization of the course, important topics to highlight, or additional academic materials. As mentioned earlier, since the LMS is not the only channel teachers can use to communicate with their students, there are two hypotheses to explain the number of teachers not demonstrating a minimum level of competence: (1) teachers use other channels such as email and in-person situations to communicate with their students; subsequently, no interaction with the LMS is recorded in this regard; and (2) in courses with more than one instructor, not all of them digitally communicate with the students via interaction with the LMS, but one of the teachers is responsible for leading this communication. Conversely, among those teachers who do interact with the LMS to communicate with the students, the majority of them (30%) achieve the maximum possible result (*Integrator B1*) in this competency, thus demonstrating their ability to interact with different tools in the LMS to communicate through different channels according to the type of message to be transmitted to the students. Again, it is worth noting that proper *netiquette* in the messages sent cannot be determined in order to preserve the anonymity of the analyzed users.

Competencies C3.2 and C3.4 obtained very similar results. These competencies are responsible for assessing the guidance or mentoring provided by the teacher and the ability to provide students with self-regulated learning tools, respectively. The vast majority of teachers (70% and 86%, respectively) do not interact with the LMS with the intention of offering guidance or mentoring to their students, nor do they employ tools to facilitate self-regulated learning. It is important to note that these results indicate that teachers do not use the LMS to offer these services, but it cannot be established whether they do so outside the interactions recorded in the LMS logs. Those teachers who do exhibit competence in these competencies predominantly present levels equivalent to *Integrator* (B1) (19% and 12%), with a few teachers reaching *Expert* (C1) level (9% and 3%).

Competency C3.3 evaluates the collaborative learning developed by students through digital technologies and virtual environments to promote interaction among students and the generation of collective knowledge. The results of this competency are distributed across two competency levels. Of the analyzed teachers, 40% do not reveal interaction with digital tools to demonstrate this competency (*Uninitiated A0*). These teachers are not using intrinsic resources in the LMS that foster collaborative learning among students. Again, this result does not imply that they are not applying it in the classroom with their students, but their interaction with the LMS does not leave records that allow for determining this skill. In contrast, 52% of teachers demonstrate a level equivalent to *Explorer* (A2), encouraging their students to work in teams by offering them specific tools provided by the LMS. A few teachers reach higher competency levels equivalent to B2 (2%) or C1 (6%), thanks to the interaction with LMS resources designed exclusively to facilitate teamwork while allowing the application of pedagogical methodologies linked to these resources.

Figure 5 shows the results obtained by the teachers after grouping the results obtained in each of the competencies analyzed. The result is calculated by adding the scores obtained in each competency and weighting them by the maximum possible. The value is then segmented into ranges that make up the final competency levels.

## 4. Discussion

Different authors support the need to assess the skills that teachers demonstrate when interacting with digital tools. While some authors propose models for assessing teachers during class situations [34,35,36,37,38,39], a different set of authors opt to use digital competency assessment frameworks [17,41,42,43,44,45,47]. 

This study has developed a model to evaluate teachers’ digital competence as defined by the DigCompEdu framework, a framework widely used to assess TDC [53,54,55,56]. The model follows a quantitative approach shared by many other proposals [40,57]. 

While the majority of instruments that assess teachers’ digital competence based on an evaluation framework are questionnaires [17,40,41,42,43,45,47,57,58], our model relies on an analysis of the data produced by teachers’ interaction with an LMS. The idea of automating the analysis of data to allow for the evaluation of large numbers of teachers has been derived from proposals put forward by other authors [34,35,36,39]. The radar charts displayed in Figure 3 are a common way of visually displaying competency levels and have also been used by other proposals [45].

Some authors express concern about the effect that teachers’ self-perception has on self-assessment models [40,43,57,59]. Self-perception tends to affect the results obtained. Our model, we believe, can bring a new point of view to teacher evaluation as it focuses on analyzing the resulting data that does not consider the teacher’s self-perception.

The results obtained are consistent with those of other authors [43,44,46,58,60]. Teachers show the best results in competency Area 2, specifically in the skills corresponding to the creation and management of digital resources, represented by competencies C2.2 and C2.3 [43,46]. Digital competence in educators ranges from basic to intermediate, showing low abilities in matters of personalizing students’ learning experiences, and standing out in the creation and management of digital resources [60]. Other authors [44,58] reach similar conclusions when determining the level of digital competence of teachers in higher education; statistical analysis shows that most educators display low to medium-low levels of digital competence. These results correlate with the instrumental competencies necessary to interact with an LMS. According to the TPACK model [27], these instrumental competencies correspond to Technological Content Knowledge (TCK); they are essential but insufficient, as they do not necessarily take into account pedagogical methodologies that facilitate teaching–learning processes. As seen in the results presented and shared by other studies, many teachers are capable of using digital instruments to create and manage digital resources but lack the expertise to reach higher competency levels by incorporating the pedagogical knowledge into their strategies. This issue can be seen again with competencies C3.1, C3.2, and C3.4, where most teachers are able to have an instrumental interaction with technology equivalent to the TCK defined by the TPACK model [27], but results cannot confirm that this interaction goes hand in hand with the pedagogical strategies employed.

The final competency level results displayed in Figure 5 draw a proficiency curve centered on the *Explorer* (A2) level, which is similar to that seen in the results of other authors [43]. Although the results obtained in other studies center the curve on the *Integrator* (B1) competence, the distribution of teachers across the competence levels is consistent with the results. One of the reasons why we believe this difference exists lies in the limitations found in this work. Not being able to assess most of the Leader C1 and Pioneer C2 competencies severely limits the ability to prove high competence levels, leading to a displacement of results.

There are significant differences with studies conducted by other authors in terms of the demonstration of competence C1.1 corresponding to Organizational Communication. While the results presented by other studies show a medium level of competence [43,46,60,61], the results obtained in our study are low. We believe the reason for the low results is that the model is not able to analyze all the communication channels used by teachers to interact with learners. The model focuses on analyzing teacher–student communication within the LMS but ignores other means of communication such as email.

## 5. Conclusions

It has been possible to successfully develop a quantitative instrument capable of automatically assessing teachers’ digital competency by means of analyzing educators’ interaction with a learning management system. Analyzing the records generated by human–computer interaction allows us to define quantitative and objective metrics. The instrument is aligned with the European framework for the digital competence of educators, DigCompEdu, which offers a reliable means of assessing digital competency. 

A proof of concept conducted with the instrument yielded satisfactory results after analyzing the interactions generated by 987 university professors across various Bachelor and Master programs, including face-to-face, hybrid, and online instruction. This underscores the feasibility and scalability of our approach, demonstrating its potential for widespread application within educational contexts.

The insights gained from the proof of concept have provided valuable foundations for identifying areas for improvement and designing targeted interventions. These findings serve as a catalyst for equipping educators with the necessary competencies to navigate the challenges of teaching in today’s knowledge society.

However, throughout the process of defining the necessary indicators, limitations were identified, highlighting the need to expand the model. While the use of an LMS proves suitable for analyzing many stages and processes related to teaching and learning, it alone is insufficient to conduct a holistic study of all competencies outlined in the DigCompEdu framework.

Therefore, future research efforts should explore additional instruments and methodologies to complement the analysis conducted within the LMS environment. Recognizing the multidimensional nature of digital competence among educators, our findings underscore the necessity of employing a diverse array of elements to comprehensively analyze each facet of digital competency.

## Figures and Tables

**Figure 1 sensors-24-03326-f001:**
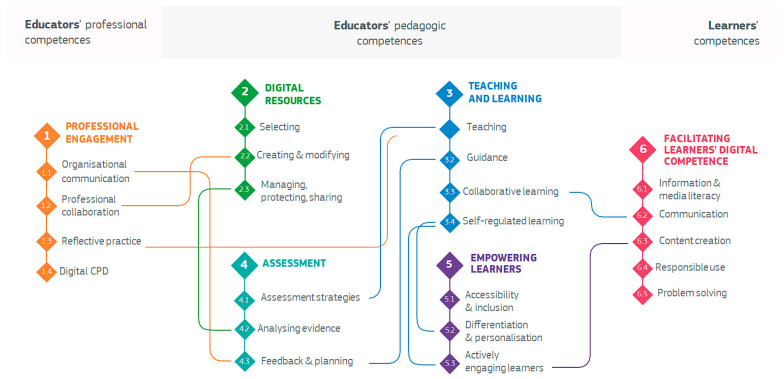
Infographic with the organization of the 22 competencies defined by DigCompEdu and organized into 6 areas [11].

**Figure 2 sensors-24-03326-f002:**
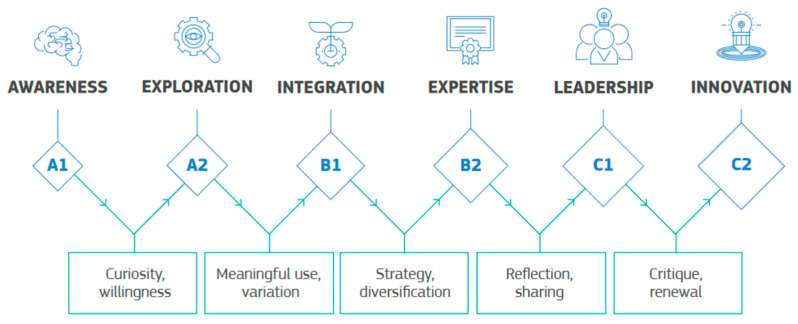
DigCompEdu progression model based on CEFR levels [11].

**Figure 3 sensors-24-03326-f003:**
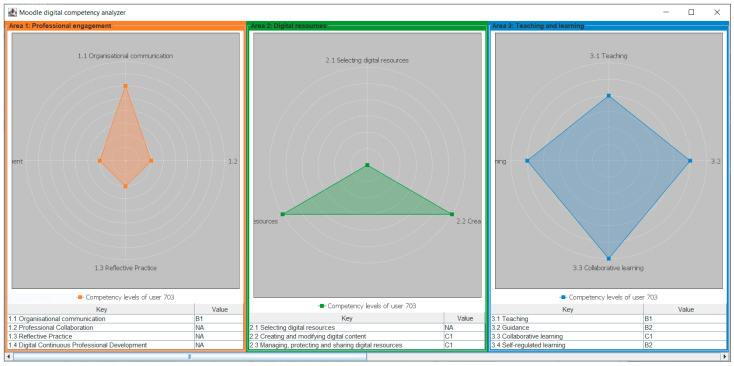
A radar-chart view of the results of the analysis of a teacher’s digital competence.

**Figure 4 sensors-24-03326-f004:**
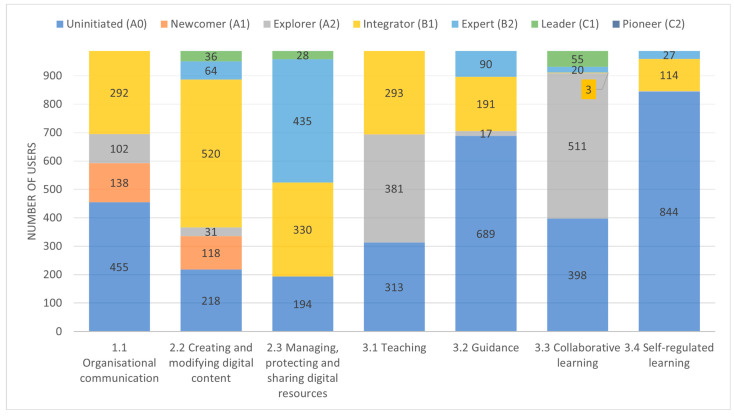
Stacked bar chart for comparative analysis of the competence levels obtained as a result of the experiment.

**Figure 5 sensors-24-03326-f005:**
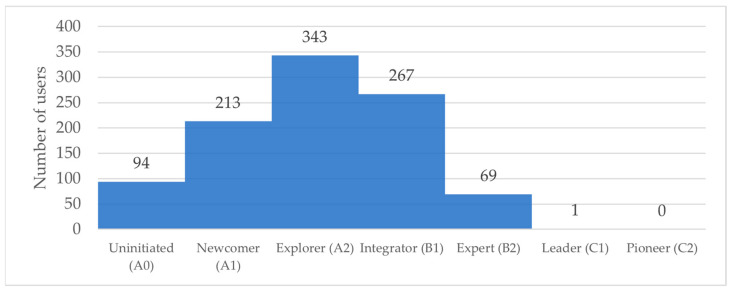
Number of users grouped by final competency level.

**Table 1 sensors-24-03326-t001:** Metrics on digital competence for core Moodle activities and resources.

Core Activity/Resource	Area 1	Area 2	Area 3	Area 4	Area 5	Area 6
(LTI) External tool	X	X	X	X	X	X
Activity completion			B2	C1		
Assignment		A1		B1		B1
Badges		B2	B2	B2	B1	
Book		B1			A2	
Chat	A2		B1			A2
Choice	B1	A2		A2		B1
Course completion			B2	C1	A2	
Course reports				C1		
Database		C1	B2	B1	B1	B2
Feedback	A2	B1		A2		B1
File		B1			A2	
Folder		B1			A2	
Forum	A2	B1	A2	B1	A2	B1
Glossary		A2	B2	B1	A2	B2
Gradebook				B1		
H5P activity		X	X	X	X	X
IMS content package		X	X	X	X	X
Label		B1			A2	
Lesson		C1	B2	B2	B2	
Page		B1			A2	
Quiz		B1	B1	B2	B1	
Restrict Access		B2		C1	B2	
Rubric	A2	B2	B2	B1		
SCORM		X	X	X	X	
Survey	B2			A2		
URL		B1			A2	
Wiki		C1	B2	B2	A2	B2
Workshop		C1	B2	B2	B1	B1

A1, A2, B1, B2, C1, and C2 is the notation defined in DigCompEdu to assess competency level [11]. X determines a possible indicator that could specify a precise level of competency.

**Table 2 sensors-24-03326-t002:** Metrics on digital competence for non-core (plugin) Moodle activities and resources.

Core Activity/Resource	Area 1	Area 2	Area 3	Area 4	Area 5	Area 6
Acknowledge plagiarism statement access rule						C1
Attendance			B2	A2		
Blackboard Collaborate	A2		A2			A2
CodeRunner		C1			B1	B2
Completion Progress			B2	C1	A2	
Configurable Reports		C1		C1		
Content Pages		B1				
Custom certificate		B1			A2	
Filter Codes		C1			B2	
Group choice	A2		A2			B1
JazzQuiz		B2	B2	B2	B2	
Level Up XP Gamification		C1	B2	B2	B1	
Multi-Language Content					B2	
Ouriginal				B2		C1
Peer Work			C1	B2	B1	B1
Questionnaire	A2	A2		A2		B1
Record audio		A2			B1	B1
Scheduler	A2			B1		
Shortcodes		C1			B2	
Structured label		B1				
Student folder		B2	B1	B1		B2
Subcourse				B1		
Teams	A2		A2			A2
Unilabel		B1			A2	
Word Select		A2			B1	
Zoom meeting	A2		A2			A2

A1, A2, B1, B2, C1, and C2 is the notation defined in DigCompEdu to assess competency level [11]. X determines a possible indicator that could specify a precise level of competency.

**Table 3 sensors-24-03326-t003:** Number of teachers who obtained a competence level in each of the competencies analyzed.

Competence Level	C1.1	C2.2	C2.3	C3.1	C3.2	C3.3	C3.4
Uninitiated (A0)	455	218	194	313	689	398	844
Newcomer (A1)	138	118	-	-	-	-	-
Explorer (A2)	102	31	-	381	17	511	2
Integrator (B1)	292	520	330	293	191	3	114
Expert (B2)	-	64	435	0	90	20	27
Leader (C1)	-	36	28	-	-	55	-
Pioneer (C2)	-	-	-	-	-	-	-

**Table 4 sensors-24-03326-t004:** Results of the statistical analysis of the data obtained on teachers’ competence levels.

Competency	µ	σ	Mo
1.1 Organizational communication	1.23	1.3	0
2.2 Creating and modifying digital content	2.2	1.44	3
2.3 Managing, protecting and sharing digital resources	2.91	1.52	4
3.1 Teaching	1.66	1.21	2
3.2 Guidance	0.98	1.52	0
3.3 Collaborative learning	1.4	1.36	2
3.4 Self-regulated learning	0.46	1.13	0

**Table 5 sensors-24-03326-t005:** Mean and Mode of the data obtained on teachers’ competence levels in DigCompEdu terms.

Competency	µ	Mo
1.1 Organizational communication	Newcomer (A1)	Uninitiated (A0)
2.2 Creating and modifying digital content	Explorer (A2)	Integrator (B1)
2.3 Managing, protecting and sharing digital resources	Integrator (B1)	Expert (B2)
3.1 Teaching	Explorer (A2)	Explorer (A2)
3.2 Guidance	Newcomer (A1)	Uninitiated (A0)
3.3 Collaborative learning	Newcomer (A1)	Explorer (A2)
3.4 Self-regulated learning	Uninitiated (A0)	Uninitiated (A0)

## Data Availability

The data that support the findings of this study are available on request from the corresponding author. The data are not publicly available due to privacy or ethical restrictions.

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
