# Peer review of "Automated Monitoring of Human–Computer Interaction for Assessing Teachers’ Digital Competence Based on LMS Data Extraction"

_sensors, 2024, doi:10.3390/s24113326_

Round 1
Reviewer 1 Report
Comments and Suggestions for Authors
1.In the Introduction part, the existing literature analysis of the evaluation framework of teachers' digital competence is relatively superficial. Please briefly state the existing literature, compare the similarities and differences of these frameworks in depth.
2.What is the correlation between the title "Teacher-computer interaction assessment" and "digital competence assessment"? Why is there "Teacher-computer interaction assessment" in the introduction? Please restate. In addition, the title "Teacher-computer interaction assessment" and the paragraph content "Finally, a third group of authors base... applied to teaching. " do not match.
3.Much of the content in the Results section belongs to the Materials and Methods section. Please readjust it.
4.The logical level of the content presented in the Results and Discussion sections are not high. It is recommended to reorganize according to the research question.
5.Please clearly highlight the 5 steps of the ADDIE methodology in the paper.
6.Please add necessary references to the Discussion section.
Comments on the Quality of English LanguagePay attention to the simplicity of the paper language and merge paragraphs with similar meanings as appropriate.
Author Response
Dear Reviewer,
Thank you very much for your time and efforts in detailing aspects for improving our proposal. Please find the detailed responses in the attachment. We hope that these changes will satisfy the weak points mentioned.

Reviewer 2 Report
Comments and Suggestions for Authors
The article discusses different assessment methods for teacher digital competence, specifically using sensors such as cameras, microphones, or electroencephalograms for recorded class video analysis. This focus on using sensors aligns well with the scope of the journal Sensors MDPI, which covers sensors and sensing technologies research.
Incorporating the sentence about the article's structure at the end of the introduction without ending with "Figure" can provide a smooth transition for the reader.
The article proposes automating the evaluation process by monitoring metrics derived from teachers’ interaction with a Learning Management System (LMS). The study successfully assessed 987 university faculty members, allowing insights into technological adoption and planning educational improvements based on individual profiles. Under the chapter Materials in the subchapter Development of the Extraction Tool, you mentioned Figure 3, but Figure 3 is under Results. Please correct this.
In the Discussion, a comparison between your research results and the findings of other studies must be included. This can be done by revisiting key sources cited in your literature review section or citing relevant studies in the discussion section. By doing so, you can either validate your results or use existing research to support your claims.
Author Response

(The authors gave the same response as above.)

Round 2
Reviewer 1 Report
Comments and Suggestions for Authors
1. In the Introduction part, it is beneficial that the paper compared the similarities and differences of these frameworks for the assessment of teacher digital competence in a new paragraph. However, the content does not sufficiently emphasize the feature of " human-computer interaction in the education sector is changing with the constant emergence of new digital technologies". It is recommended to highlight the novel aspect of human-computer interaction for the assessment frameworks in the new paragraph from line 128 to line 138.
2. It is recommended that the Introduction section clearly state the research questions. Then, in the Results and Discussion section, the paper should address and respond to each research question in separate paragraphs dedicated to each question.
3. Is Teacher Digital Competence and Teachers’ Digital Competence the same term? Please ensure consistency throughout the paper. Additionally, "Teachers’ Digital Competence" or "Teacher Digital Competence" should not be treated as a proper noun, and it is recommended to use lowercase letters accordingly.
Comments on the Quality of English Language
Pay attention to the consistency of word presentation.
Author Response
Dear Reviewer,
Thank you very much again for your time and efforts in this second review. Please find the detailed responses below.
- In the Introduction part, it is beneficial that the paper compared the similarities and differences of these frameworks for the assessment of teacher digital competence in a new paragraph. However, the content does not sufficiently emphasize the feature of " human-computer interaction in the education sector is changing with the constant emergence of new digital technologies". It is recommended to highlight the novel aspect of human-computer interaction for the assessment frameworks in the new paragraph from line 128 to line 138.
We recognize that the way in which the paragraph has been written may generate confusion to the reader, because the frameworks do not reflect the evolution of digital technology and its necessary interaction with humans. The frameworks focus on determining that the use of technology in education is vital in a digital world, without going into detail about what type of technology (and their interaction with it) is involved.
To clarify this evolution in technology and its interaction. A new paragraph in line 51 has been included to exemplify the changes and possibilities that digital technology has brought to education.
In addition, the paragraph in line 151 has been updated.
- It is recommended that the Introduction section clearly state the research questions. Then, in the Results and Discussion section, the paper should address and respond to each research question in separate paragraphs dedicated to each question.
The research question has been added in line 160, at the end of the subsection reviewing the frameworks. As we only have one research question, it is not possible to organize the Results and Discussion section based on multiple RQ. For the results section, our intention is to first present the details of each competency in order from best to worst results, and finally, to analyse the overall results. For the Discussion, we have tried to follow the suggestion made to us by another reviewer. First, we compare our model with the rest of the models found in the literature. Then, we wanted to validate our results by comparing them with the results obtained with the rest of the models and tools. At this point we have followed the same criterion of first presenting the results where teachers tend to stand out, and then highlighting the points for improvement. Finally, we would like to dedicate the final paragraph of our discussion to highlight the most significant differences found when analysing teachers with our model, based on the results of other authors.
- Is Teacher Digital Competence and Teachers’ Digital Competence the same term? Please ensure consistency throughout the paper. Additionally, "Teachers’ Digital Competence" or "Teacher Digital Competence" should not be treated as a proper noun, and it is recommended to use lowercase letters accordingly.
The term teacher’s digital competence has been reviewed for consistency. The literature cites both “Teacher Digital Competence” and “Teachers’ Digital Competence”. Given that the term represents the digital competence of teachers, it was decided to use the latter form. The term has been transformed to lowercase, as recommended by the reviewer. The initial intention in capitalizing the term was to be able to treat the term and its acronym TDC interchangeably. In the end, however, it was decided to use the term for the most part in order to avoid an abuse of the acronym that might make it difficult to understand for readers who are not familiar with it.